# Variational Memory Addressing in Generative Models

**Jörg Bornschein**  **Andriy Mnih**  **Daniel Zoran**  **Danilo J. Rezende**
{bornschein, amnih, danielzoran, danilor}@google.com
DeepMind, London, UK

## Abstract

Aiming to augment generative models with external memory, we interpret the output of a memory module with stochastic addressing as a conditional mixture distribution, where a read operation corresponds to sampling a discrete memory address and retrieving the corresponding content from memory. This perspective allows us to apply variational inference to memory addressing, which enables effective training of the memory module by using the target information to guide memory lookups. Stochastic addressing is particularly well-suited for generative models as it naturally encourages multimodality which is a prominent aspect of most high-dimensional datasets. Treating the chosen address as a latent variable also allows us to quantify the amount of information gained with a memory lookup and measure the contribution of the memory module to the generative process. To illustrate the advantages of this approach we incorporate it into a variational autoencoder and apply the resulting model to the task of generative few-shot learning. The intuition behind this architecture is that the memory module can pick a relevant template from memory and the continuous part of the model can concentrate on modeling remaining variations. We demonstrate empirically that our model is able to identify and access the relevant memory contents even with hundreds of unseen Omniglot characters in memory.

## 1  Introduction

Recent years have seen rapid developments in generative modelling. Much of the progress was driven by the use of powerful neural networks to parameterize conditional distributions composed to define the generative process (e.g., VAEs [1, 2], GANs [3]). In the Variational Autoencoder (VAE) framework for example, we typically define a generative model $p(\mathbf{z})$, $p_\theta(\mathbf{x}|\mathbf{z})$ and an approximate inference model $q_\phi(\mathbf{z}|\mathbf{x})$. All conditional distributions are parameterized by multilayered perceptrons (MLPs) which, in the simplest case, output the mean and the diagonal variance of a Normal distribution given the conditioning variables. We then optimize a variational lower bound to learn the generative model for $\mathbf{x}$. Considering recent progress, we now have the theory and the tools to train powerful, potentially non-factorial *parametric* conditional distributions $p(\mathbf{x}|\mathbf{y})$ that generalize well with respect to $\mathbf{x}$ (normalizing flows [4], inverse autoregressive flows [5], etc.).

Another line of work which has been gaining popularity recently is memory augmented neural networks [6, 7, 8]. In this family of models the network is augmented with a memory buffer which allows read and write operations and is persistent in time. Such models usually handle input and output to the memory buffer using differentiable "soft" write/read operations to allow back-propagating gradients during training.

Here we propose a memory-augmented generative model that uses a discrete latent variable $a$ acting as an address into the memory buffer $\mathbf{M}$. This stochastic perspective allows us to introduce a variational approximation over the addressing variable which takes advantage of target information

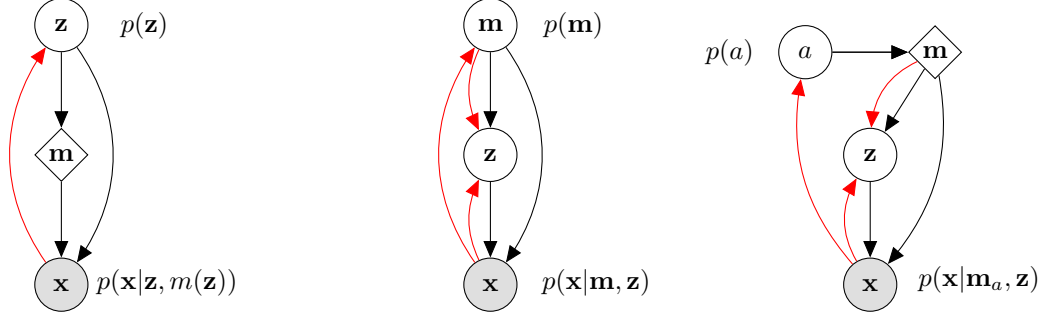

Figure 1: **Left**: Sketch of typical SOTA generative latent variable model with memory. Red edges indicate approximate inference distributions $q(\cdot|\cdot)$. The $KL(q||p)$ cost to identify a specific memory entry might be substantial, even though the cost of accessing a memory entry should be in the order of $\log|\mathbf{M}|$. **Middle & Right**: We combine a top-level categorical distribution $p(a)$ and a conditional variational autoencoder with a Gaussian $p(\mathbf{z}|\mathbf{m})$.

when retrieving contents from memory during training. We compute the sampling distribution over the addresses based on a learned similarity measure between the memory contents at each address and the target. The memory contents $\mathbf{m}_a$ at the selected address $a$ serve as a context for a continuous latent variable $\mathbf{z}$, which together with $\mathbf{m}_a$ is used to generate the target observation. We therefore interpret memory as a non-parametric conditional mixture distribution. It is non-parametric in the sense that we can change the content and the size of the memory from one evaluation of the model to another without having to relearn the model parameters. And since the retrieved content $\mathbf{m}_a$ is dependent on the stochastic variable $a$, which is part of the generative model, we can directly use it downstream to generate the observation $\mathbf{x}$. These two properties set our model apart from other work on VAEs with mixture priors [9, 10] aimed at unconditional density modelling. Another distinguishing feature of our approach is that we perform sampling-based variational inference on the mixing variable instead of integrating it out as is done in prior work, which is essential for scaling to a large number of memory addresses.

Most existing memory-augmented generative models use soft attention with the weights dependent on the continuous latent variable to access the memory. This does not provide clean separation between inferring the address to access in memory and the latent factors of variation that account for the variability of the observation relative to the memory contents (see Figure 1). Or, alternatively, when the attention weights depend deterministically on the encoder, the retrieved memory content can not be directly used in the decoder.

Our contributions in this paper are threefold: a) We interpret memory-read operations as conditional mixture distribution and use amortized variational inference for training; b) demonstrate that we can combine discrete memory addressing variables with continuous latent variables to build powerful models for generative few-shot learning that scale gracefully with the number of items in memory; and c) demonstrate that the KL divergence over the discrete variable $a$ serves as a useful measure to monitor memory usage during inference and training.

## 2 Model and Training

We will now describe the proposed model along with the variational inference procedure we use to train it. The generative model has the form

$$p(\mathbf{x}|\mathbf{M}) = \sum_a p(a|\mathbf{M}) \int_{\mathbf{z}} p(\mathbf{z}|\mathbf{m}_a)\, p(\mathbf{x}|\mathbf{z}, \mathbf{m}_a)\, d\mathbf{z} \tag{1}$$

where $\mathbf{x}$ is the observation we wish to model, $a$ is the addressing categorical latent variable, $\mathbf{z}$ the continuous latent vector, $\mathbf{M}$ the memory buffer and $\mathbf{m}_a$ the memory contents at the $a$th address.

The generative process proceeds by first sampling an address $a$ from the categorical distribution $p(a|\mathbf{M})$, retrieving the contents $\mathbf{m}_a$ from the memory buffer $\mathbf{M}$, and then sampling the observation $x$ from a conditional variational auto-encoder with $\mathbf{m}_a$ as the context conditioned on (Figure 1, B).

The intuition here is that if the memory buffer contains a set of templates, a trained model of this type should be able to produce observations by distorting a template retrieved from a randomly sampled memory location using the conditional variational autoencoder to account for the remaining variability.

We can write the variational lower bound for the model in (1):

$$\log p(\mathbf{x}|\mathbf{M}) \geq \mathbb{E}_{a,\mathbf{z}\sim q(\cdot|\mathbf{M},\mathbf{x})} \left[\log p(\mathbf{x},\mathbf{z},a|\mathbf{M}) - \log q(a,\mathbf{z}|\mathbf{M},\mathbf{x})\right] \tag{2}$$

$$\text{where } q(a,\mathbf{z}|\mathbf{M},\mathbf{x}) = q(a|\mathbf{M},\mathbf{x})q(\mathbf{z}|\mathbf{m}_a,\mathbf{x}). \tag{3}$$

In the rest of the paper, we omit the dependence on $\mathbf{M}$ for brevity. We will now describe the components of the model and the variational posterior (3) in detail.

The first component of the model is the memory buffer $\mathbf{M}$. We here do not implement an explicit write operation but consider two possible sources for the memory content: **Learned memory:** In generative experiments aimed at better understanding the model's behaviour we treat $\mathbf{M}$ as model parameters. That is we initialize $\mathbf{M}$ randomly and update its values using the gradient of the objective. **Few-shot learning:** In the generative few-shot learning experiments, before processing each minibatch, we sample $|\mathbf{M}|$ entries from the training data and store them in their raw (pixel) form in $\mathbf{M}$. We ensure that the training minibatch $\{\mathbf{x}_1, ..., \mathbf{x}_{|\mathcal{B}|}\}$ contains disjoint samples from the same character classes, so that the model can use $\mathbf{M}$ to find suitable templates for each target $\mathbf{x}$.

The second component is the addressing variable $a \in \{1, ..., |\mathbf{M}|\}$ which selects a memory entry $\mathbf{m}_a$ from the memory buffer $\mathbf{M}$. The varitional posterior distribution $q(a|\mathbf{x})$ is parameterized as a softmax over a similarity measure between $\mathbf{x}$ and each of the memory entries $\mathbf{m}_a$:

$$q_\phi(a|\mathbf{x}) \propto \exp \mathrm{S}_\phi^q(\mathbf{m}_a, \mathbf{x}), \tag{4}$$

where $\mathrm{S}_\phi^q(\mathbf{x}, \mathbf{y})$ is a learned similarity function described in more detail below.

Given a sample $a$ from the posterior $q_\phi(a|\mathbf{x})$, retreiving $\mathbf{m}_a$ from $M$ is a purely deterministic operation. Sampling from $q(a|\mathbf{x})$ is easy as it amounts to computing its value for each slot in memory and sampling from the resulting categorical distribution. Given $a$, we can compute the probability of drawing that address under the prior $p(a)$. We here use a learned prior $p(a)$ that shares some parameters with $q(a|\mathbf{x})$.

**Similarity functions:** To obtain an efficient implementation for mini-batch training we use the same memory content $\mathbf{M}$ for the all training examples in a mini-batch and choose a specific form for the similarity function. We parameterize $\mathrm{S}^q(\mathbf{m}, \mathbf{x})$ with two MLPs: $\mathrm{h}_\phi$ that embeds the memory content into the matching space and $\mathrm{h}_\phi^q$ that does the same to the query $\mathbf{x}$. The similarity is then computed as the inner product of the embeddings, normalized by the norm of the memory content embedding:

$$\mathrm{S}^q(\mathbf{m}_a, \mathbf{x}) = \frac{\langle \mathbf{e}_a, \mathbf{e}^q \rangle}{||\mathbf{e}_a||_2} \tag{5}$$

$$\text{where } \mathbf{e}_a = \mathrm{h}_\phi(\mathbf{m}_a) \ , \ \mathbf{e}^q = \mathrm{h}_\phi^q(\mathbf{x}). \tag{6}$$

This form allows us to compute the similarities between the embeddings of a mini-batch of $|\mathcal{B}|$ observations and $|\mathbf{M}|$ memory entries at the computational cost of $O(|\mathbf{M}||\mathcal{B}||\mathbf{e}|)$, where $|\mathbf{e}|$ is the dimensionality of the embedding. We also experimented with several alternative similarity functions such as the plain inner product ($\langle \mathbf{e}_a, \mathbf{e}^q \rangle$) and the cosine similarity ($\langle \mathbf{e}_a, \mathbf{e}^q \rangle / ||\mathbf{e}_a|| \cdot ||\mathbf{e}^q||$) and found that they did not outperform the above similarity function. For the unconditioneal prior $p(a)$, we learn a query point $\mathbf{e}^p \in \mathcal{R}^{|e|}$ to use in similarity function (5) in place of $\mathbf{e}^q$. We share $\mathrm{h}_\phi$ between $p(a)$ and $q(a|\mathbf{x})$. Using a trainable $p(a)$ allows the model to learn that some memory entries are more useful for generating new targets than others. Control experiments showed that there is only a very small degradation in performance when we assume a flat prior $p(a) = 1/|\mathbf{M}|$.

## 2.1 Gradients and Training

For the continuous variable $\mathbf{z}$ we use the methods developed in the context of variational autoencoders [1]. We use a conditional Gaussian prior $p(z|\mathbf{m}_a)$ and an approximate conditional posterior $q(z|\mathbf{x}, \mathbf{m}_a)$. However, since we have a discrete latent variable $a$ in the model we can not simply backpropagate gradients through it. Here we show how to use VIMCO [11] to estimate the gradients for

this model. With VIMCO, we essentially optimize the multi-sample variational bound [12, 13, 11]:

$$\log p(\mathbf{x}) \geq \mathop{\mathbb{E}}_{\substack{a^{(k)} \sim q(a|\mathbf{x}) \\ \mathbf{z}^{(k)} \sim q(\mathbf{z}|\mathbf{m}_a, \mathbf{x})}} \left[ \log \frac{1}{K} \sum_{k=1}^{K} \frac{p(\mathbf{x}, \mathbf{m}_a, \mathbf{z})}{q(a, \mathbf{z}|\mathbf{x})} \right] = \mathcal{L} \tag{7}$$

Multiple samples from the posterior enable VIMCO to estimate low-variance gradients for those parameters $\phi$ of the model which influence the non-differentiable discrete variable $a$. The corresponding gradient estimates are:

$$\nabla_\theta \mathcal{L} \simeq \sum_{a^{(k)}, z^{(k)} \sim q(\cdot|\mathbf{x})} \omega^{(k)} \left( \nabla_\theta \log p_\theta(\mathbf{x}, a^{(k)}, \mathbf{z}^{(k)}) - \nabla_\theta \log q_\theta(\mathbf{z}|a, \mathbf{x}) \right) \tag{8}$$

$$\nabla_\phi \mathcal{L} \simeq \sum_{a^{(k)}, z^{(k)} \sim q(\cdot|\mathbf{x})} \omega_\phi^{(k)} \nabla_\phi \log q_\phi(a^{(k)}|\mathbf{x})$$

$$\text{with } \omega^{(k)} = \frac{\tilde{\omega}^{(k)}}{\sum_k \tilde{\omega}^{(k)}} \ , \ \tilde{\omega}^{(k)} = \frac{p(\mathbf{x}, a^{(k)}, \mathbf{z}^{(k)})}{q(a^{(k)}, \mathbf{z}^{(k)}|\mathbf{x})}$$

$$\text{and } \omega_\phi^{(k)} = \log \frac{1}{K} \sum_{k'} \tilde{\omega}^{(k')} - \log \frac{1}{K-1} \sum_{k' \neq k} \tilde{\omega}^{(k')} - \omega^{(k)}$$

For **z**-related gradients this is equivalent to IWAE [13]. Alternative gradient estimators for discrete latent variable models (e.g. NVIL [14], RWS [12] or Gumbel-max relaxation-based approaches [15, 16]) might work here too, but we have not investigated their effectiveness. Notice how the gradients $\nabla \log p(\mathbf{x}|\mathbf{z}, a)$ provide updates for the memory contents $\mathbf{m}_a$ (if necessary), while the gradients $\nabla \log p(a)$ and $\nabla \log q(a|\mathbf{x})$ provide updates for the embedding MLPs. The former update the mixture components while the latter update their relative weights. The log-likelihood bound (2) suggests that we can decompose the overall loss into three terms: the expected reconstruction error $\mathbb{E}_{a, \mathbf{z} \sim q} [\log p(x|a, \mathbf{z})]$ and the two KL terms which measure the information flow from the approximate posterior to the generative model for our latent variables: $KL(q(a|\mathbf{x})||p(a))$, and $\mathbb{E}_{a \sim q} [KL(q(\mathbf{z}|a, \mathbf{x})||p(\mathbf{z}|a))]$.

## 3 Related work

Attention and external memory are two closely related techniques that have recently become important building blocks for neural models. Attention has been widely used for supervised learning tasks such as translation, image classification and image captioning. External memory can be seen as an input or an internal state and attention mechanisms can either be used for selective reading or incremental updating. While most work involving memory and attention has been done in the context supervised learning, here we are interested in using them effectively in the generative setting.

In [17] the authors use soft-attention with learned memory contents to augment models to have more parameters in the generative model. External memory as a way of implementing one-shot generalization was introduced in [18]. This was achieved by treating the exemplars conditioned on as memory entries accessed through a soft attention mechanism at each step of the incremental generative process similar to the one in DRAW [19]. Generative Matching Networks [20] are a similar architecture which uses a single-step VAE generative process instead of an iterative DRAW-like one. In both cases, soft attention is used to access the exemplar memory, with the address weights computed based on a learned similarity function between an observation at the address and a function of the latent state of the generative model.

In contrast to this kind of deterministic soft addressing, we use hard attention, which stochastically picks a single memory entry and thus might be more appropriate in the few-shot setting. As the memory location is stochastic in our model, we perform variational inference over it, which has not been done for memory addressing in a generative model before. A similar approach has however been used for training stochastic attention for image captioning [21]. In the context of memory, hard attention has been used in RLNTM – a version of the Neural Turing Machine modified to use stochastic hard addressing [22]. However, RLNTM has been trained using REINFORCE rather than variational inference. A number of architectures for VAEs augmented with mixture priors have

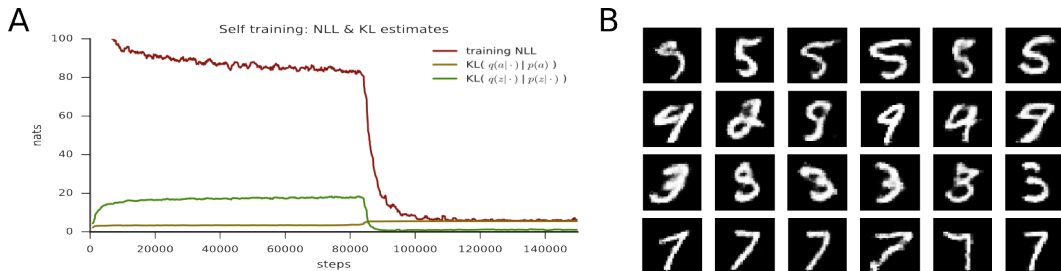

Figure 2: **A:** Typical learning curve when training a model to recall MNIST digits ($\mathbf{M} \sim$ training data (each step); $\mathbf{x} \sim \mathbf{M}$; $|\mathbf{M}| = 256$): In the beginning the continuous latent variables model most of the variability of the data; after $\approx 100k$ update steps the stochastic memory component takes over and both the NLL bound and the $\mathrm{KL}(q(a|\mathbf{x})||p(a))$ estimate approach $\log(256)$, the NLL of an optimal probabilistic lookup-table. **B:** Randomly selected samples from the MNIST model with learned memory: Samples within the same row use a common $\mathbf{m}_a$.

been proposed, but they do not use the mixture component indicator variable to index memory and integrate out the variable instead [9, 10], which prevents them from scaling to a large number of mixing components.

An alternative approach to generative few-shot learning proposed in [23] uses a hierarchical VAE to model a large number of small related datasets jointly. The statistical structure common to observations in the same dataset are modelled by a continuous latent vector shared among all such observations. Unlike our model, this model is not memory-based and does not use any form of attention. Generative models with memory have also been proposed for sequence modelling in [24], using differentiable soft addressing. Our approach to stochastic addressing is sufficiently general to be applicable in this setting as well and it would be interesting how it would perform as a plug-in replacement for soft addressing.

## 4 Experiments

We optimize the parameters with Adam [25] and report experiments with the best results from learning rates in {1e-4, 3e-4}. We use minibatches of size 32 and $K$=4 samples from the approximate posterior $q(\cdot|\mathbf{x})$ to compute the gradients, the KL estimates, and the log-likelihood bounds. We keep the architectures deliberately simple and do not use autoregressive connections or IAF [5] in our models as we are primarily interested in the quantitative and qualitative behaviour of the memory component.

### 4.1 MNIST with fully connected MLPs

We first perform a series of experiments on the binarized MNIST dataset [26]. We use 2 layered en- and decoders with 256 and 128 hidden units with ReLU nonlinearities and a 32 dimensional Gaussian latent variable $\mathbf{z}$.

**Train to recall:** To investigate the model's capability to use its memory to its full extent, we consider the case where it is trained to maximize the likelihood for random data points $\mathbf{x}$ which are present in $\mathbf{M}$. During inference, an optimal model would pick the template $\mathbf{m}_a$ that is equivalent to $\mathbf{x}$ with probability $q(a|\mathbf{x})$=1. The corresponding prior probability would be $p(a) \approx {}^1/|\mathbf{M}|$. Because there are no further variations that need to be modeled by $\mathbf{z}$, its posterior $q(\mathbf{z}|\mathbf{x}, \mathbf{m})$ can match the prior $p(\mathbf{z}|\mathbf{m})$, yielding a KL cost of zero. The model expected log likelihood would be -$\log |\mathbf{M}|$, equal to the log-likelihood of an optimal probabilistic lookup table. Figure 2A illustrates that our model converges to the optimal solution. We observed that the time to convergence depends on the size of the memory and with $|\mathbf{M}| > 512$ the model sometimes fails to find the optimal solution. It is noteworthy that the trained model from Figure 2A can handle much larger memory sizes at test time, e.g. achieving NLL $\approx \log(2048)$ given 2048 test set images in memory. This indicates that the matching MLPs for $q(a|\mathbf{x})$ are sufficiently discriminative.

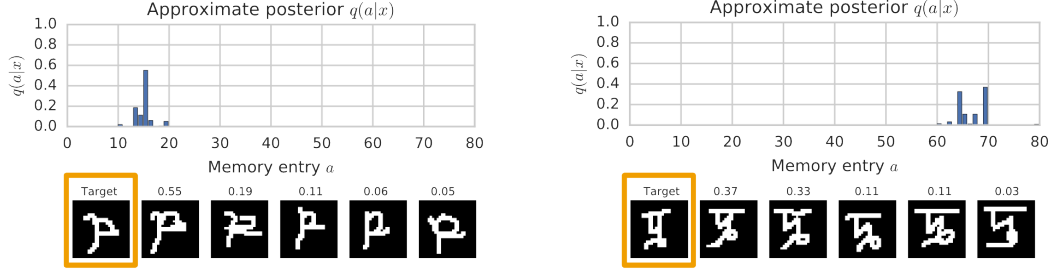

Figure 3: Approximate inference with $q(a|\mathbf{x})$: Histogram and corresponding top-5 entries $\mathbf{m}_a$ for two randomly selected targets. $\mathbf{M}$ contains 10 examples from 8 unseen test-set character classes.

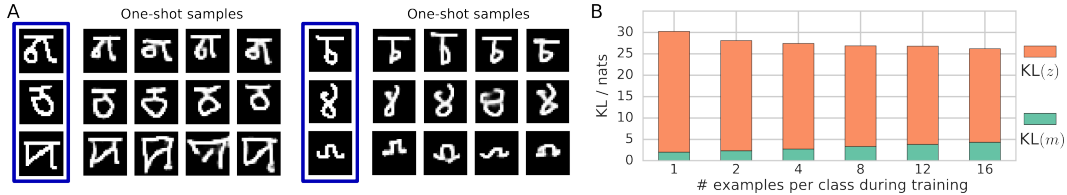

Figure 4: **A:** Generative one-shot sampling: Left most column is the testset example provided in $\mathbf{M}$; remaining columns show randomly selected samples from $p(\mathbf{x}|\mathbf{M})$. The model was trained with 4 examples from 8 classes each per gradient step. **B:** Breakdown of the KL cost for different models trained with varying number of examples per class in memory. $KL(q(a|\mathbf{x})||p(a))$ increases from 2.0 to 4.5 nats as $KL(q(\mathbf{z}|\mathbf{m}_a, \mathbf{x})||p(\mathbf{z}|\mathbf{m}_a))$ decreases from 28.2 to 21.8 nats. As the number of examples per class increases, the model shifts the responsibility for modeling the data from the continuous variable $\mathbf{z}$ to the discrete $a$. The overall testset NLL for the different models improves from 75.1 to 69.1 nats.

**Learned memory**: We train models with $|\mathbf{M}| \in \{64, 128, 256, 512, 1024\}$ randomly initialized mixture components ($\mathbf{m}_a \in \mathcal{R}^{256}$). After training, all models converged to an average $KL(q(a|\mathbf{x})||p(a)) \approx 2.5 \pm 0.3$ nats over both the training and the test set, suggesting that the model identified between $e^{2.2} \approx 9$ and $e^{2.8} \approx 16$ clusters in the data that are represented by $a$. The entropy of $p(a)$ is significantly higher, indicating that multiple $\mathbf{m}_a$ are used to represent the same data clusters. A manual inspection of the $q(a|\mathbf{x})$ histograms confirms this interpretation. Although our model overfits slightly more to the training set, we do generally not observe a big difference between our model and the corresponding baseline VAE (a VAE with the same architecture, but without the top level mixture distribution) in terms of the final NLL. This is probably not surprising, because MNIST provides many training examples describing a relatively simple data manifold. Figure 2B shows samples from the model.

## 4.2 Omniglot with convolutional MLPs

To apply the model to a more challenging dataset and to use it for generative few-shot learning, we train it on various versions of the Omniglot [27] dataset. For these experiments we use convolutional en- and decoders: The approximate posterior $q(\mathbf{z}|\mathbf{m}, \mathbf{x})$ takes the concatenation of $\mathbf{x}$ and $\mathbf{m}$ as input and predicts the mean and variance for the 64 dimensional $\mathbf{z}$. It consists of 6 convolutional layers with $3 \times 3$ kernels and 48 or 64 feature maps each. Every second layer uses a stride of 2 to get an overall downsampling of $8 \times 8$. The convolutional pyramid is followed by a fully-connected MLP with 1 hidden layer and $2|\mathbf{z}|$ output units. The architecture of $p(\mathbf{x}|\mathbf{m}, \mathbf{z})$ uses the same downscaling pyramid to map $\mathbf{m}$ to a $|\mathbf{z}|$-dimensional vector, which is concatenated with $\mathbf{z}$ and upscaled with transposed convolutions to the full image size again. We use skip connections from the downscaling layers of $\mathbf{m}$ to the corresponding upscaling layers to preserve a high bandwidth path from $\mathbf{m}$ to $\mathbf{x}$. To reduce overfitting, given the relatively small size of the Omniglot dataset, we tie the parameters of the convolutional downscaling layers in $q(\mathbf{z}|\mathbf{m})$ and $p(\mathbf{x}|\mathbf{m}, \mathbf{z})$. The embedding MLPs for $p(a)$ and $q(a|\mathbf{x})$ use the same convolutional architecture and map images $\mathbf{x}$ and memory content $\mathbf{m}_a$ into

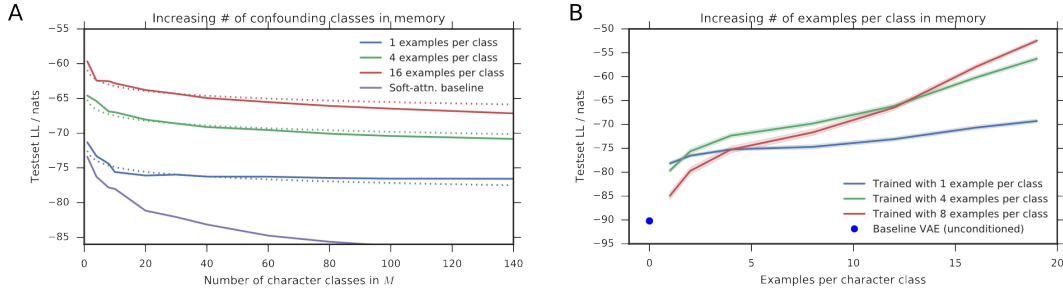

Figure 5: Robustness to increasing memory size at test-time: **A:** Varying the number of confounding memory entries: At test-time we vary the number of classes in $\mathbf{M}$. For an optimal model of disjoint data from C classes we expect $\mathcal{L} = $ average $\mathcal{L}$ per class $+ \log C$ (dashed lines). The model was trained with 4 examples from 8 character classes in memory per gradient step. We also show our best soft-attenttion baseline model which was trained with 16 examples from two classes each gradient step. **B:** Memory contains examples from all 144 test-set character classes and we vary the number of examples per class. At C=0 we show the LL of our best unconditioned baseline VAE. The models were trained with 8 character classes and $\{1, 4, 8\}$ examples per class in memory.

a 128-dimensional matching space for the similarity calculations. We left their parameters untied because we did not observe any improvement nor degradation of performance when tying them.

**With learned memory:** We run experiments on the $28 \times 28$ pixel sized version of Omniglot which was introduced in [13]. The dataset contains 24,345 unlabeled examples in the training, and 8,070 examples in the test set from 1623 different character classes. The goal of this experiment is to show that our architecture can learn to use the top-level memory to model highly multi-modal input data. We run experiments with up to 2048 randomly initialized mixture components and observe that the model makes substantial use of them: The average $KL(q(a|\mathbf{x})||p(a))$ typically approaches $\log |\mathbf{M}|$, while $KL(q(\mathbf{z}|\cdot)||p(\mathbf{z}|\cdot))$ and the overall training-set NLL are significantly lower compared to the corresponding baseline VAE. However big models without regularization tend to overfit heavily (e.g. training-set NLL < 80 nats; testset NLL > 150 nats when using $|\mathbf{M}|$=2048). By constraining the model size ($|\mathbf{M}|$=256, convolutions with 32 feature maps) and adding 3e-4 L2 weight decay to all parameters with the exception of $\mathbf{M}$, we obtain a model with a testset NLL of 103.6 nats (evaluated with K=5000 samples from the posterior), which is about the same as a two-layer IWAE and slightly worse than the best RBMs (103.4 and $\approx$100 respectively, [13]).

**Few-shot learning:** The $28 \times 28$ pixel version [13] of Omniglot does not contain any alphabet or character-class labels. For few-shot learning we therefore start from the original dataset [27] and scale the $104 \times 104$ pixel sized examples with $4 \times 4$ max-pooling to $26 \times 26$ pixels. We here use the 45/5 split introduced in [18] because we are mostly interested in the quantitative behaviour of the memory component, and not so much in finding optimal regularization hyperparameters to maximize performance on small datasets. For each gradient step, we sample 8 random character-classes from random alphabets. From each character-class we sample 4 examples and use them as targets $\mathbf{x}$ to form a minibatch of size 32. Depending on the experiment, we select a certain number of the remaining examples from the same character classes to populate $\mathbf{M}$. We chose 8 character-classes and 4 examples per class for computational convenience (to obtain reasonable minibatch and memory sizes). In control experiments with 32 character classes per minibatch we obtain almost indistinguishable learning dynamics and results.

To establish meaningful baselines, we train additional models with identical encoder and decoder architectures: 1) A simple, unconditioned VAE. 2) A memory-augmented generative model with soft-attention. Because the soft-attention weights have to depend solely on the variables in the generative model and may not take input directly from the encoder, we have to use $\mathbf{z}$ as the top-level latent variable: $p(\mathbf{z}), p(\mathbf{x}|\mathbf{z}, \mathbf{m}(\mathbf{z}))$ and $q(\mathbf{z}|\mathbf{x})$. The overall structure of this model resembles the structure of prior work on memory-augmented generative models (see section 3 and Figure 1A), and is very similar to the one used in [20], for example.

For the unconditioned baseline VAE we obtain a NLL of 90.8, while our memory augmented model reaches up to 68.8 nats. Figure 5 shows the scaling properties of our model when varying the number of conditioning examples at test-time. We observe only minimal degradation compared

| Model | $C_{test}$ | 1 | 2 | 3 | 4 | 5 | 10 | 19 |
|---|---|---|---|---|---|---|---|---|
| Generative Matching Nets | 1 | 83.3 | 78.9 | 75.7 | 72.9 | 70.1 | 59.9 | 45.8 |
| Generative Matching Nets | 2 | 86.4 | 84.9 | 82.4 | 81.0 | 78.8 | 71.4 | 61.2 |
| Generative Matching Nets | 4 | 88.3 | 87.3 | 86.7 | 85.4 | 84.0 | 80.2 | 73.7 |
| Variational Memory Addressing | 1 | 86.5 | 83.0 | 79.6 | 79.0 | 76.5 | 76.2 | 73.9 |
| Variational Memory Addressing | 2 | 87.2 | 83.3 | 80.9 | 79.3 | 79.1 | 77.0 | 75.0 |
| Variational Memory Addressing | 4 | 87.5 | 83.3 | 81.2 | 80.7 | 79.5 | 78.6 | 76.7 |
| Variational Memory Addressing | 16 | 89.6 | 85.1 | 81.5 | 81.9 | 81.3 | 79.8 | 77.0 |

Table 1: Our model compared to Generative Matching Networks [20]: GMNs have an extra stage that computes joint statistics over the memory context which gives the model a clear advantage when multiple conditiong examples per class are available. But with increasing number of classes $C$ it quickly degrades. LL bounds were evaluated with $K$=1000 posterior samples.

to a theoretically optimal model when we increase the number of concurrent character classes in memory up to 144, indicating that memory readout works reliably with $|\mathbf{M}| \geq 2500$ items in memory. The soft-attention baseline model reaches up to 73.4 nats when $\mathbf{M}$ contains 16 examples from 1 or 2 character-classes, but degrades rapidly with increasing number of confounding classes (see Figure 5A). Figure 3 shows histograms and samples from $q(a|\mathbf{x})$, visually confirming that our model performs reliable approximate inference over the memory locations.

We also train a model on the Omniglot dataset used in [20]. This split provides a relatively small training set. We reduce the number of feature channels and hidden layers in our MLPs and add 3e-4 L2 weight decay to all parameters to reduce overfitting. The model in [20] has a clear advantage when many examples from very few character classes are in memory because it was specifically designed to extract joint statistics from memory before applying the soft-attention readout. But like our own soft-attention baseline, it quickly degrades as the number of concurrent classes in memory is increased to 4 (table 1).

**Few-shot classification:** Although this is not the main aim of this paper, we can use the trained model to perform discriminative few-shot classification: We can estimate $p(c|x) \approx \sum_{\mathbf{m}_a \text{ has label } c} \mathbb{E}_{\mathbf{z} \sim q(\mathbf{z}|a, \mathbf{x})} [p(\mathbf{x}, \mathbf{z}, \mathbf{m}_a)/p(\mathbf{x})]$ or by using the feed forward approximation $p(c|x) \approx \sum_{\mathbf{m}_a \text{ has label } c} q(a|\mathbf{x})$. Without any further retraining or finetuneing we obtain classification accuracies of 91%, 97%, 77% and 90% for 5-way 1-shot, 5-way 5-shot, 20-way 1-shot and 20-way 5-shot respectively with $q(a|\mathbf{x})$.

## 5 Conclusions

In our experiments we generally observe that the proposed model is very well behaved: we never used temperature annealing for the categorical softmax or other tricks to encourage the model to use memory. The interplay between $p(a)$ and $q(a|x)$ maintains exploration (high entropy) during the early phase of training and decreases naturally as the sampled $\mathbf{m}_a$ become more informative. The KL divergences for the continuous and discrete latent variables show intuitively interpretable results for all our experiments: On the densely sampled MNIST dataset only a few distinctive mixture components are identified, while on the more disjoint and sparsely sampled Omniglot dataset the model chooses to use many more memory entries and uses the continuous latent variables less. By interpreting memory addressing as a stochastic operation, we gain the ability to apply a variational approximation which helps the model to perform precise memory lookups during inference and training. Compared to soft-attention approaches, we loose the ability to naively backprop through read-operations and we have to use approximations like VIMCO. However, our experiments strongly suggest that this can be a worthwhile trade-off. Our experiments also show that the proposed variational approximation is robust to increasing memory sizes: A model trained with 32 items in memory performed nearly optimally with more than 2500 items in memory at test-time. Beginning with $\mathbf{M} \geq 48$ our hard-attention implementation becomes noticeably faster in terms of wall-clock time per parameter update than the corresponding soft-attention baseline. Even though we use $K$=4 posterior samples during training and the soft-attention baseline only requires a single one.

**Acknowledgments**

We thank our colleagues at DeepMind and especially Oriol Vinyals and Sergey Bartunov for insightful discussions.

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
