[Reviews · NeurIPS 2017]

Reviewer 1



The authors propose variational memory addressing. It augments generative models with external memory and hard attention, and interestingly, derives read and write mechanisms that mimick more classical probabilistic graphical models than the more sophisticated mechanisms as in, e.g., neural Turing machines. In their formulation, external memory acts much like a global variable in topic models and mixture models, whether they sample a "membership" given by the hard attention and proceed to generate the local variable z and data x conditional on memory indexed by this membership. I found this a particularly useful way of understanding memory in the context of latent variable models, where writing corresponds to inference. As the authors note in, e.g., L175-186, it seems the algorithm does not scale well with respect to the external memory size. This can be justified mathematically as the the variance of the black box gradients with respect q(a) parameters increases with the size of a. It is unlikely that VIMCO can help much in this regard. That said, I'm impressed that the authors were able to get interesting results with |M| up to 1024.

Reviewer 2



This paper is not the first one that addresses variational autoencoders with external memory modules (which the authors falsely claim). For instance, Chongxuan Li, Jun Zhu, Bo Zhang, "Learning to Generate with Memory," Proc. ICML 2016 solve the same problem, in a match more potent way than this work (I reckon this method as a quite trivial extension of existing VAEs). Since the authors have failed to identify all the related work, and properly compare to them, this paper certainly cannot be accepted to NIPS.

Reviewer 3



This paper proposes a memory-augmented generative model that performs stochastic, discrete memory addressing, by interpreting the memory as a non-parametric conditional mixture distribution. This variational memory addressing model can combine discrete memory addressing variables with continuous latent variables, to generate samples only with few samples in the memory, which is useful for few-shot learning. The authors implement a VAE version of their model and validate it for the few-shot recognition tasks on the Omniglot dataset, on which it significantly outperforms the Generative Matching Networks, which is an existing memory-augmented network model. Further analysis shows that the proposed model accesses relevant part of the memory even with hundreds of unseen instances in the memory. Pros: - Performing discrete, stochastic memory addressing for memory-augmented generative model is a novel idea which makes sense. Also, the authors have done a good job in motivating why this model is superior to soft attention approach. - The proposed variational addressing scheme is shown to work well in case of few-shot learning, even in case where existing soft-attention model fails to work. - The proposed scheme of interpreting the memory usage with KL divergence seems useful. Cons -Not much, except that the experimental study only considers character data, although they are standard datasets. It would be better if the paper provides experimental results on other types of data, such as images, and compared against (or coupled with) recent generative models (such as GANs) Overall, this is a good paper that presents a novel, working idea. It has effectively solved the problem with existing soft-attention memory addressing model, and is shown to work well for few-shot learning. Thus I vote for accepting the paper. - Some typos: Line 104: unconditioneal -> unconditional Line 135: "of" is missing between context and supervised. Line 185: eachieving -> achieving Table 1 is missing a label "number of shots" for the numbers 1,2,3,4,5,10 and 19.